# Governance and Corporate Management System Supported by Innovation, Technology, and Digital Transformation as a Driver of Change

Alexander Guerrero-Avendaño [1,*], Wilson Nieto Bernal [2] and Carmenza Luna Amaya [3]

1   Facultad de Ingeniería, Universidad Francisco de Paula Santander, Ocaña 546551, Colombia
2   Departamento de Ingeniería de Sistemas, Universidad del Norte, Barranquilla 081007, Colombia; wnieto@uninorte.edu.co
3   Departamento de Ingeniería Industrial, Universidad del Norte, Barranquilla 081007, Colombia; cluna@uninorte.edu.co
*   Correspondence: aguerreroav@ufpso.edu.co

**Abstract:** Governance and Corporate Management Systems have become key tools for the comprehensive management of organizations, which, when implemented, guarantee corporate success, which can materialize in obtaining corporate goals, strategic objectives, KPIs, and KRP. This paper is the product of research exercises where a Governance and Corporate Management System is exposed, supported by three main pillars: Technology Management, Innovation Management, and Digital Transformation, which we will call GCMS+ from now on. The work developed is based on a quasi-experimental longitudinal research methodology, in which three key phases are addressed; the first phase corresponds to the systematic review of the literature, which allows the identification of the key components of the proposed GCMS+ system. The second phase of the methodology focused on the modelling of the GCMS+ system, which visually and descriptively describes the structural components, flows, and control mechanisms. The third phase of the methodology focuses on the validation of the GCMS+ system proposed from the discussed literature and contrasting the position of the authors, which is identified through a conceptual discussion. In this case, it seeks to validate the structural elements of GCMS+, the constraints, adoption of good practices, integration of standards, adoption of principles, integration of drivers, strategic taxonomy, organizational structures, governance and management approach, as well as methodologies, roles, initiatives, metrics, and indicators.

**Keywords:** technology management; innovation management; strategy; digital transformation; organisational transformation; corporate government; enterprise architecture

## 1. Introduction

In recent years, research has emerged on organisational change management [1], innovation processes and models [2–7], the use of resources to increase sustainability [8,9] and to assess the growth capacity of organisations [1,10,11]. The model presented in this study has a corporate approach, which seeks to improve the response capacity to the global market.

Organisations are constantly challenged to innovate in different areas, not to fail, and to remain competitive [12–14]; they resort to the implementation of several types of models, such as Technology Management models that generate value through the effective use of technology [2,15,16], Innovation Management models for the creation of new knowledge and generation of ideas to develop new or improved products, processes, and services [17,18], Digital Transformation models facilitate the reorganisation of processes to increase digital capabilities and generate more excellent value for customer [19–21], and

Organisational Transformation Models that creates changes in business vision and impact the ability to compete [22], creating actions that allow innovation to be generated [23].

These models and processes allow organisations to learn about global trends in technology, corporate governance and reduce risk by strengthening their capabilities, optimising their resources, generating new knowledge and managing their commercial, manufacturing, and service operations more effectively. All of this is possible once the challenges of adopting new business models and collaborative strategies are overcome [24].

As previously mentioned, the use of new technologies has its complications; emerging economic markets are affected by product Innovation [14,25]; emerging economies and their measurement variables are constantly changing, which requires ensuring the growth and projection of the sector [26], by estimating its evolution through an economic model.

With the incorporation of appropriate management systems, organisations can continually monitor and, thus, more easily standardise their activities; management systems take advantage of the knowledge of their environment and make it the main asset to devise their technology. It must be understood that management systems and platforms are different from each other [12], and prototypes must be designed for each type of organisation [27]. For best results, management systems should be incorporated into the construction of strategic plans to align system implementation with projected objectives and ensure greater success during development [28].

This study proposes to design a conceptual model of Governance and Corporate Management System (GCMS+) that integrates Technology Management, Innovation Management and Digital Transformation. The creation of a new business model configuration with transparency as a principle, combining digital management with sustainability guidelines to develop strategic and sustainable innovation capabilities, is proposed.

The structure of this article begins with a review of the literature on the concept of corporate governance and the most representative models. Through these initial readings, the concept of a Governance and Corporate Management System and its elements are explored in depth. The Materials and Methods section details the data collection and analysis stage, the bibliometric analysis, and its components. The fourth section details the system principles and components and describes the flow of the system.

## 2. Theoretical Background

### 2.1. Corporate Governance

Corporate Governance (CG) refers to how organisations are administered, managed, and controlled. Many studies have focused on ensuring transparency, accountability, equity, and sustainability in the functioning of organisations [29–32], ensuring that leaders and managers act in accordance with the organisation's strategic goals and the best interests of all stakeholders [29].

Some authors indicate that corporate governance is an essential factor for the success and sustainable performance of an organisation regardless of its size [33,34]; they also suggest that effective and quality governance can improve the productivity of organisations, provide an enabling environment for sustainable success, improve management efficiency, and reduce the adverse effects of economic shocks [35].

Wahyuningrum et al. define Corporate Governance as the set of activities and rules for the proper functioning of organisations in terms of the processes by which organisations are directed and controlled [30], while Du Plessis et al. define Corporate Governance as the structure of rules, practices and processes used to run and manage a business, which essentially involves balancing the interests of the many public and private stakeholders in the organisation [36].

Internationally, legislation has been issued to support corporate governance that seeks to provide guidelines to reduce risks to shareholders, employees, and consumers, such as the UK Cardbury Report [37] in 1992, the Corporations Act of Australia [38] in 2001, and the USA Sarbanes–Oxley Act [39] in 2002.

Among the organisations that have issued a concept of corporate governance and provided a framework for the application of good governance is the Organisation for Economic Co-operation and Development (OECD), which states that corporate governance refers to a set of interactions and relationships that occur between different actors within an organisation "the management of the organisation, its board of directors, its shareholders and other interested parties" [40], in 1999. The OECD first published principles for corporate governance were reviewed and approved by the G20 leaders in 2015, and, in its most recent version (2023), they have been designated by the Financial Stability Board (FSB) as one of its standards [41].

### 2.2. Corporate Governance Models

Several studies have attempted to explain the different styles of corporate governance through models [42]; the Anglo-American model, also known as the Anglo-Saxon model, is based on the idea that the shareholders are the owners of the organisation [43] and have the right to control management and make essential decisions [44]. This model is the basis of corporate governance in countries such as the United States, Australia, Great Britain, and Canada. Among the advantages attributed to this model are strong protection of shareholders rights and high transparency in financial disclosure [45], which indicates that there must be a strict enforcement of disclosure rules and ensures that shareholders have access to relevant information to make informed decisions and be able to effectively control the management of the organisation [46]. The disadvantages of the Anglo-American model are its focus on short-term profit and its disregard for minority stakeholders, which can lead to short-term decisions that are not beneficial to the organisation in the long term.

The Japanese corporate governance model changed significantly after the bubble crisis of the 1990s, influenced by the 1999 and 2015 OECD Principles of Corporate Governance [40], where at least one independent external director was introduced on the board of directors of organisations [47], increasing internal control and the active participation of strategic shareholders in the management process to stimulate economic efficiency and harmonise the interests of the organisation's partners and employees [48]. This model facilitates the flexible supervision and financing of organisations, as well as effective communication between them and banks, which are the main source of financing [49]. The Japanese model seeks to provide the best benefit to shareholders (who can also be creditors, suppliers, or customers), which encourages long-term decision-making and promotes the sustainability of the organisation [49]. A disadvantage of the model is that in some organisations, the funders are in senior management positions, which limits the decision-making power of shareholders [50], leading to a possible lack of transparency and a decrease in investor confidence.

The German corporate governance model is characterised by practices of separation of management and control, allowing shareholders and employees to make crucial decisions in the organisation in order to control its management [51]. Significant shareholders actively participate in the management process to stimulate economic efficiency; this model features a more robust and active monitoring approach than the Anglo-American model [46,52]. The German governance system focuses on the influence of strong capital markets and not on internal control, as in the Japanese model [47].

In Latin America, there is no standard model of corporate governance; however, the importance of developing balanced internal control and corporate governance structures is highlighted [53], and each country adapts according to its business culture and regulatory framework [54]. In Mexico, the corporate governance model is based on the general shareholders' meeting, which appoints the review of financial statements, emphasising the independence of the board of directors and the protection of shareholders' rights, which brings it closer to the Anglo-American model [42]. On the other hand, the Brazilian corporate governance model combines elements of the Anglo-American model with greater stakeholder participation and the adoption of social inclusion policies [55]. In Colombia, the Code of Good Corporate Governance is the basis of the corporate governance model and

focuses on transparency, social responsibility, and the active participation of the different governing bodies in strategic decision-making and oversight of corporate management, in addition to the fact that the board of directors must form an audit committee and a nomination committee [55].

## 3. Materials and Methods

This study seeks to determine the elements to build a Governance and Corporate Management System, for which a systematic review of the literature was used, analysing articles covering Corporate Governance, Corporate Management, Technology Management, Innovation Management, Digital Transformation, and Organisational Transformation. The article describes the processes involved in the model. It presents a schematic diagram of the processes and the feedback developed with the flow, as well as the interactions with the environment and the systems that support capacity enhancement.

### 3.1. Data Collection

The review was conducted primarily as a structured search for keywords and terms such as "Corporate Management", "Corporate Governance", "Organisational Transformation", "Digital Transformation", "Innovation Management", and "Technology Management", in combination with "system", "innovation", "strategy", "business", and "Model". The search for articles was carried out mainly in the Web of Science bibliographic database between 2000 and 2022, filtered by type of document selecting only articles and reviews in English and Spanish; the result was organised by the most cited.

To approach the systematic literature review, a characterisation of the search terms was carried out using as a guide a study on research and Technology Management [56] and bibliometrix, a tool in R for the comprehensive analysis of scientific mapping [57].

Terms were searched across titles, abstracts, and keywords to obtain several articles focusing on corporate governance and management.

### 3.2. Data Download

Four data samples were downloaded: (1) Corporate Governance with 263 observations, (2) Technology Management, (3) Innovation Management, and (4) Digital Transformation.

The articles resulting from the search were downloaded in BibTex (bib) format from the Web of Science database, with all author information, affiliations, citations, abstracts, keywords, and bibliographic information.

The data was then prepared for mapping analysis using the bibliometrix library, version 3.0.2 in RStudio 1.3.959 for macOS.

### 3.3. Bibliometric Analysis of the Data

For the analysis of this study, bibliometrics was first used to analyse the characteristics of the literature as an object of research; critical items (country, citation, keyword) were analysed quantitatively using statistical and mathematical methods [57]. The sample group performed the result of the bibliometric mapping analysis; relevant results are shown in Table 1 and Figure 1 below.

For the Corporate Governance sample, 358 documents were collected, with the USA being the leader in article publication, while China ranks second. The United States, Italy, and the United Kingdom had the most citations. In this sample, the keyword plus (repeated terms from the titles of the articles cited in the bibliography) most used in the articles were Performance and Ownership.

Of the 387 documents in the Technology Management sample, China had the highest number of articles published, and the USA was in second place. For the countries with the most citations, the results showed the following: USA first, then China, and Germany. Performance, Impact, and Adoption were the most used keywords in this sample.

**Table 1.** Statistical details on publications 2000–2022.

| Year | 2000 | 2001 | 2002 | 2003 | 2004 | 2005 | 2006 | 2007 | 2008 | 2009 | 2010 | 2011 |
|---|---|---|---|---|---|---|---|---|---|---|---|---|
| Corporate Governance | | 4 | 8 | 3 | 5 | 8 | 13 | 12 | 18 | 16 | 16 | 21 |
| Technology Management | 1 | 7 | 7 | 5 | 8 | 5 | 7 | 7 | 11 | 13 | 10 | 15 |
| Innovation Management | | 5 | 10 | 8 | 5 | 9 | 18 | 13 | 16 | 26 | 29 | 23 |
| Digital Transformation | | | | 2 | | 1 | 1 | | | | | 2 |
| **Year** | **2012** | **2013** | **2014** | **2015** | **2016** | **2017** | **2018** | **2019** | **2020** | **2021** | **2022** | **Total** |
| Corporate Governance | 11 | 9 | 16 | 15 | 16 | 6 | 31 | 32 | 27 | 30 | 41 | 358 |
| Technology Management | 13 | 16 | 15 | 5 | 9 | 11 | 32 | 30 | 22 | 50 | 88 | 387 |
| Innovation Management | 24 | 28 | 35 | 34 | 35 | 20 | 42 | 47 | 34 | 28 | 9 | 498 |
| Digital Transformation | 2 | 1 | | 2 | 3 | 1 | 11 | 20 | 33 | 62 | 105 | 246 |

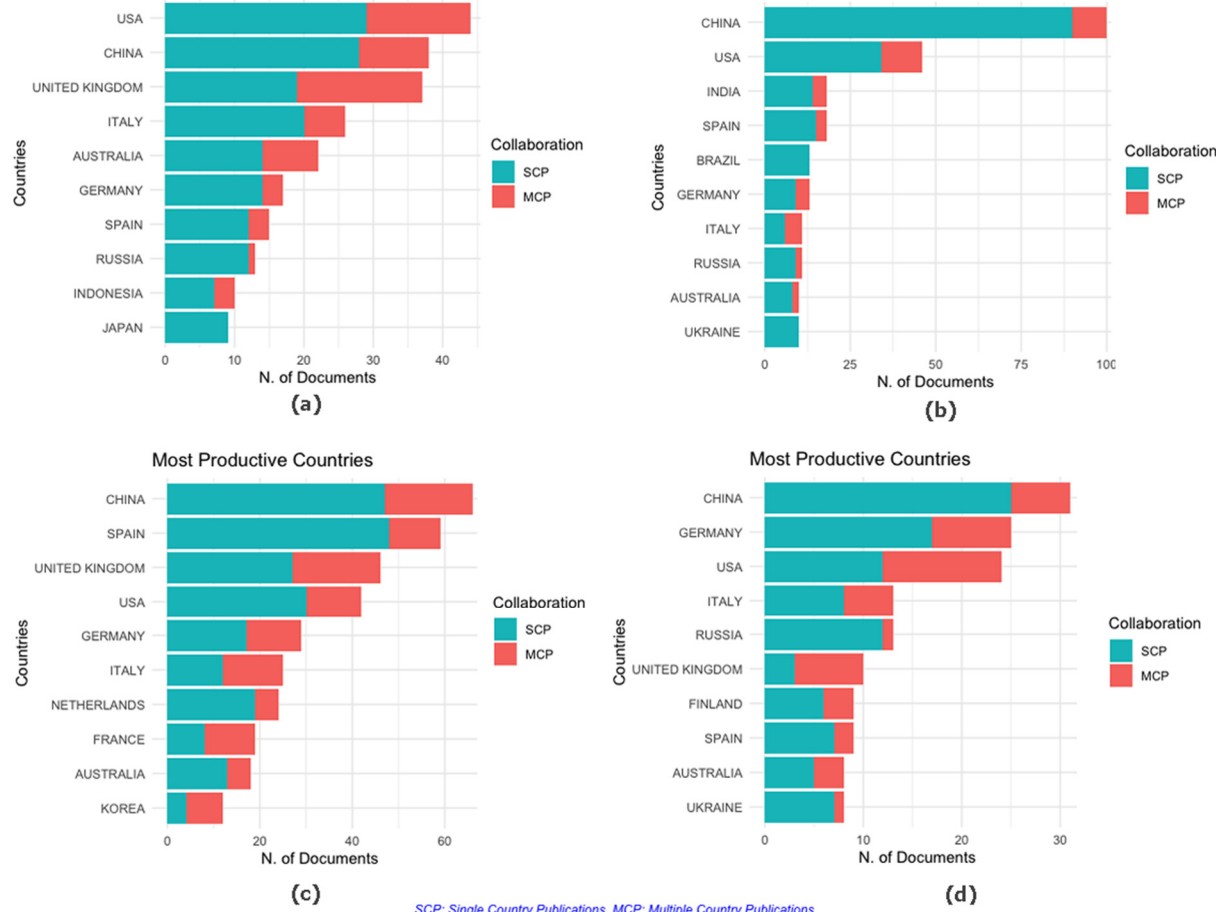

**Figure 1.** Bibliometric analysis of countries with the most publications on (**a**) Corporate Governance, (**b**) Technology Management, (**c**) Innovation Management, and (**d**) Digital Transformation.

Finally, the sample on Digital Transformation was made up of 246 documents, with China, Germany, and the USA being the countries with the highest number of article publications, and Germany and the USA with the most cited articles. Management, Innovation, Technology, and Performance were the most used keywords in the articles.

The results of the sample on Innovation Management, in which 498 documents were used, revealed that the countries with the highest number of article publications were China, Spain, the United Kingdom, and the United States. From the sample, the countries with the highest number of citations were the USA, United Kingdom, China, and Spain, and the most used plus keywords in the articles were Performance, Impact, Research and Development, and Capabilities.

*3.4. Content Analysis*

In this study, the content analysis complemented the bibliometric analysis to identify and characterise the elements and extract knowledge about the existing research studies [56] on Corporate Governance, Technology Management, Innovation Management, and Digital Transformation. The authors of this study conducted a detailed content analysis of the articles they considered to be the most influential in identifying the preliminary elements that would make up the Governance and Corporate Management System.

After the content analysis, the preliminary list of elements was confronted through an expert judgment to discuss the relevance and functionality of each of the elements [58,59]; as a result of the analysis and the expert judgement, Table 2 lists the elements and items analysed to make up the system.

**Table 2.** Relevant articles for the construction of the Governance and Corporate Management System.

| Capacities | Elements | Authors |
|---|---|---|
| Technology management | Strategic Direction, Collaboration Networks. | [60,61] |
| | Policies. | [62–64] |
| | Impact metrics. | [65] |
| | Planning and Management System. | [66–69] |
| Innovation management | Motivations and Resources. | [13,26,70–73] |
| | Functional dynamics. | [10,64,74,75] |
| | Organisational routines. | [76,77] |
| | Framework, Roles, Standards, Policies. | [63,73,78–82] |
| Digital Transformation | Environments, Partnerships, Capabilities. | [83–86] |
| | Risk mitigation tools. | [87,88] |
| | Framework, Roles, Standards, Policies, Research. | [12,14,86,89–95] |
| | IT Resources, Social networks. | [96–98] |
| Organisational Transformation | Entrepreneurship, Interface Design. | [23,99,100] |
| | Process Change, Organisational Culture. | [99,101–106] |
| | Business and Government Environment Technologies. | [23,102] |
| | Communication Strategy. | [103,104,107–111] |

### 3.5. Validation Using the Expert Judgement Technique

The expert judgement technique was implemented to confirm the validity of the conceptual model of the Governance and Corporate Management System, its components and elements [112]. Expert judgement is a validation method increasingly used in research [113] and serves to verify the reliability of the study through the informed opinion of qualified experts who can provide information, judgements, and assessments [114]; for the expert judgement, the support of eight professionals with a master's degree and more than ten years of experience in at least one of the components of the system was requested.

As a result of the expert judgement, experts have affirmed that the model is acceptable. However, they have recommended describing the components and the flow of the system details. They suggest these adjustments are essential to ensure the model faithfully represents the system and its behaviour. The requested adjustments are detailed below.

### 3.6. Governance and Corporate Management System

Governance aims to find an efficient way to manage an organisation and optimise the resources used in each activity. Governance refers to an optimal relationship or interaction of procedures, policies, instruments, and the organisation that allows the protection of knowledge to take place.

For Deloitte, the concept of Corporate Governance refers to the set of principles, procedures and standards that regulate the functioning of the organisation's governance areas [115].

The concept of governance came into use some decades ago in European countries, Canada, the United States, and Australia as a consequence of the need of minority shareholders of an organisation to know the state of the company in which they had invested and to know what the future expectations of the company were [116].

It has been mentioned that governance is vital for organisations in general to achieve success [117], and the best strategy to achieve this is through good corporate governance practices; as a solution, instruments were developed to generate trust; in this way, corporate governance management is currently an essential criterion in organisations, with the generation of value being the main asset to be protected. The OECD states that corporate governance encompasses a set of relationships between the management of the organisation, its board of directors, shareholders, and other stakeholders [40].

Organisations that apply good governance practices thrive on reliability and security in their processes and operations as well as in their products and services, thus, increasing their level of recognition. Although implementing good governance practices does not guarantee the success of organisations, it does have a positively impact on the value and image of the brand, thus, facilitating access to foreign markets and external investment.

### 3.7. Technology Management as a Component of the Governance and Corporate Management System

Technology Management (TM) is conceived as the set of complex and multidimensional processes or activities aimed at the acquisition, incorporation, optimisation, control, learning, dissemination, and continuous improvement of the technology necessary for the execution of the processes developed in the organisation [62,118–120]. It also involves the innovation processes through Research and Development (R&D) of both product and process technologies, as well as those used in management functions [61], and is, therefore, a vital determinant of technological change for the success or failure of organisations [60,121].

Since technological change is a dynamic process [75] which constantly creates new challenges and opportunities for new products, services, processes, and organisational development [69], a frequent analysis of the characteristics that affect innovation is necessary [71], in which the links among competitive advantages [60], competitive priorities, and organisational competencies are understood [66,121]; a transformation is then required that integrates technological change with strategic aspects in an innovation system. In con-

clusion, understanding technological change requires expert guidance [62,67] that guides Technology Management towards a dynamic innovation system approach [75].

However, Technology Management is conceived as the set of techniques enabling an organisation to develop and implement its innovation and improvement plans to maintain or increase its competitive position [122]; therefore, although with the implementation of Technology Management, some organisations have not been able to increase the capabilities of their organisations in terms of innovation and improvement [65] and satisfaction rates [60,68], other studies suggest that Technology Management is the recommended way to protect and combine the strategic orientation, processes, human, technical and financial resource structure, and the management of the technology and the environment [61,123] in each activity [69] to meet the objectives of the organisation.

Technology Management also allows a better articulation between research, industry, and society through strategies introduced in the strategic plan for technological development [124] through inventorying, monitoring, evaluating, enriching, optimising, and protecting [125].

### 3.8. Innovation Management as a Component of the Governance and Corporate Management System

The set of activities aimed at the application of knowledge for the introduction of a new device, method, or material into a product or process that improves its characteristics, capabilities, or performance [73,81] is known as Technological Innovation; however, if the innovation developed also increases organisational productivity through increased sales volume of new products, thus gaining market acceptance, it is known as Innovation Management (IM).

Innovation Management is therefore an efficient combination of change management and innovation management [63,73] and the management of innovation processes [76]; the former aims to promote the process of generating new knowledge while maintaining a culture of sustainable innovation [77], and the second is incorporated into change management, acquiring a greater capacity to adapt and anticipate or provoke ruptures, to improve competitive advantages based on of the management coverage level.

Innovation Management as a system allows us to evaluate the impact of interactions of elements of the environment in the field of interest and to manage innovation [26]; it is necessary to analyse the elements that make up the innovation system, which must then be articulated according to its level of effectiveness in the development of innovation [71] and its integration with the dynamics of the system environment [64,73]. The above is to establish those elements that allow the business environment to increase its technological, learning, innovation, strategic management, and resource management capabilities [64,65].

The International Standards Organisation ISO, in its ISO 56000 standard, is a guide for the use of tools and methods "For innovation partnership", the standard describes guidelines and criteria for generating value by developing joint activities; the document covers product, business process, marketing and organisational innovation and provides a structured approach for organisations seeking to enter into an innovation partnership with another organisation [126].

Innovation Management and Technology Management in the business environment can share the stages of creation or acquisition, diffusion, and use of technology when it comes to the development of technological innovations; however, it is possible to develop innovations outside of technology when these are not technological or refer to organisational or strategic aspects. Furthermore, in innovation, the use of knowledge beyond the industrial application considers the introduction and diffusion or acceptance in the market of new or improved products, processes, business models, or organisational or marketing methods, as they are technological innovations that depend on advances in knowledge [76].

### 3.9. Digital Transformation as a Component of the Governance and Corporate Management System

Digital Transformation (DT) has led industries to face a fundamental change associated with the rapid emergence of digital technology applications [91,127,128] in all aspects.

Digital Transformation is the application of digital capabilities across processes, products and assets to improve efficiency, enhance customer value [83], manage risk [87] and discover new revenue-generating opportunities.

Digital Transformation rebuilds the dynamics of organisations to adapt them to the needs of the present and the future; a definition of Digital Transformation, associated with this dynamic of change [94], is the one mentioned by Vial (2019), where he states that digital technologies create disruptions that trigger strategic responses for value creation while managing structural changes [86].

Digital Transformation is seen as the final stage and the new beginning in the adoption of digital technologies [85], which starts with the acquisition of the competencies required to use digital technology (digital literacy), which will then be implemented and used to finally solve the challenges of organisations [129].

With the implementation of Digital Transformation strategies in organisations to generate changes in the culture of innovation [94], resources are better distributed to modernise the customer experience [12,94], creating connections between organisations [85] and new business models through the incorporation of digital technologies [83] and encouraging the use of social networks [97] to examine organisational relationships [98] and obtain more innovative and agile responses [130,131] in real time [92]. In conclusion, Digital Transformation strategies offer more efficient solutions [93] that contribute to business performance [83,132].

Digital Transformation is already being implemented at the government level; in Europe, the German Research Foundation (DFG) approved a project entitled "*Taking digital transformation to the next level*", an innovative information infrastructure for research [90].

### 3.10. Organisational Transformation as a Component of the Governance and Corporate Management System

For Hellriegel, Jackson, and Solcum (2005), organisational change refers to the transformation associated with the design or operation of an organisation [133]; this change requires planning processes that involve Organisational Transformation with changes based on the business vision, actions that permeate the culture, and actions that permeate the culture of the organisation [101] and the ability to compete [22], employing simultaneous and exploration strategies [108], actions to evaluate its performance, variations in its structure [104], and constantly test new ways of generating innovation [23].

As previously stated by the authors, Organisational Transformation always implies changes that will require the organisation to adjust to the demands of the environment, for which it must define future strategies and create a long-term vision to increase its competitiveness. With this clear picture, organisations that are involved in transformation processes seek to optimise the use of resources and adjust for the specialisation of functions reflected in the organisation's structure.

Organisations engaged in a competitive struggle [23], an attitude of renewal and permanent learning [100], have always managed to innovate in the right way, and the success of obtaining this result is due to the adoption of Organisational Transformation (OT) [102].

Wischnesvky (2004) states that Organisational Transformation is a novel change in organisational dimensions such as strategy, structure, and systems [134].

The scientific literature related to Organisational Transformation processes indicates that there are generally four critical stages within the process: the initial definition of the vision or ultimate goal, followed by the identification of the competencies required to meet the goal, the definition of strategies, budget and implementation plans, and finally the assessment of the availability of the necessary resources, according to whose feasibility the implementation of the strategies begins. Thus, achieving a planned organisational transformation with a synergetic and holistic transformation framework will depend on the monitoring of the stages described above.

## 4. Conceptual Model of the Governance and Corporate Management System

Figure 2 represents the Governance and Corporate Management System, elaborated under the innovation processes and systems approach, which is made up of elements and the relationships between them, belonging to the productive, financial, scientific, and technological environments within an institutional, legal, and cultural framework, typical of the interaction area of the Governance and Corporate Management Systems.

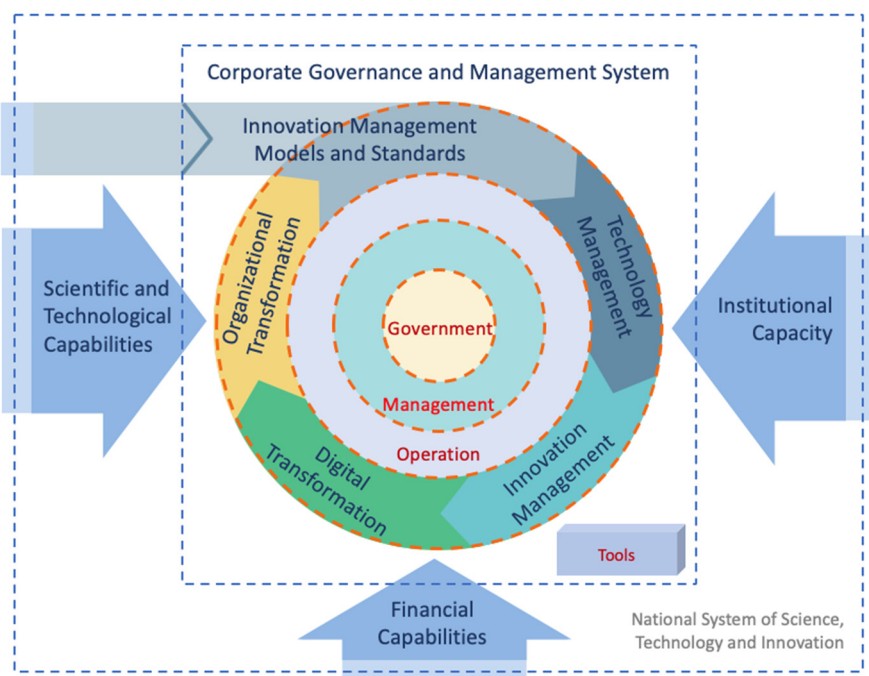

**Figure 2.** Proposed Governance and Corporate Management System.

The principles, components, description, and flow of the Governance and Corporate Management System are specified below:

### 4.1. Principles of the System

The proposed Governance and Corporate Management System model has the following special features:

- Holistic: The model incorporates human, process, and technology perspectives linked to management and corporate governance across strategic, operational, and organisational areas.
- Integrated: applies the paradigms of reliability and value for money to integrate standards and common elements from the core of management and corporate governance, from which emanates the set of policies that impact management and operation, covering all levels and organisational processes and defining a single ethical framework of organisational behaviour.
- Transparency: by integrating digital management with sustainability guidelines in corporate governance, the system allows the substitution of administrative behaviour controls for the validation of the service by different users, favouring not only the transparency of corporate and organisational management but also the execution of processes and procedures reliably, making effective use of economic and financial information.

### 4.2. Components

The *Governance and Corporate Management System* model, in addition to the Technological Management, Innovation Management and Digital Transformation components mentioned above, consists of a set of components or support tools, such as (a) business intelligence applications required by corporate governance in the functions of strategic

direction, Audit and Senior Management; (b) an information system composed of a core system and resource planning systems that support corporate management; (c) information systems with sub systems, modules, and components to support the different operational and primary activities; (d) IoT, Big Data, machine learning to facilitate Digital Transformation, applications for planning, and agile methods to support Innovation Management, and applications for technological management and Operation Technologies to support Technology Management.

*4.3. System Description*

The proposed model developed under the corporate approach of innovation processes and systems, can integrate and complement the processes of Technology Management, Innovation Management, Digital Transformation, and Organisational Transformation around the patterns of governance, management, and operation at the core of the organisations, whose moderating input is given by the models and standards of Innovation Management and the organisational scientific and technological capabilities, the financial capabilities and the corporate capacity of the environment, these inputs are given by the elements of the innovation system of the scenario, in which the organisations interact.

During the literature review, the main capacities suggested in previous studies were analysed to select the processes that are part of the proposed system, with the respective elements configuring them (see Table 2). Based on the critical processes identified, the system is constructed under the paradigm of interaction and adaptation to the externalities of the economic environment through processes of organisational transformation.

This model stipulated that innovation standards and management should be aligned with the regulations governing the development of innovation activities, the Innovation Management support system, and all existing governmental instruments.

The main contribution of the model is to facilitate the identification of groups of actors, according to the organisational characteristics and capacities that must be applied in the form of interaction to activate the value chain.

It is essential to highlight that the principle of transparency proposes to integrate digital management with sustainability guidelines in corporate governance, thus allowing it to permeate all areas of the organisation, decision-making, risk management, and business strategy; another essential point is that, at a higher level, it is possible to periodically evaluate the performance of the system and make effective use of economic-financial information.

The different capacities of the environment represented in the system may or may not be part of the geographical, political, and technological limits of the organisations, with whose elements relationships or interactions can be established for the acquisition of technologies and knowledge that promote management and transformation processes, based on internal capacities that support such interactions or through intermediaries or corporate partners that guarantee the achievement of strategic objectives.

*4.4. Flows of the Governance and Corporate Management System*

The flow of the Governance and Corporate Management System starts from its core in the Governance pattern, radiating to the entire organisation the regulatory framework, policies, strategic, operational, and monitoring roles for proper control and direction to ensure transparency, integration, sustainability, and a holistic approach in the governance pattern; it is defined as putting each of the components into operation, and the necessary parameters for the delimitation of capabilities are obtained. The plans, objectives, and strategies are specified in the manual of functions and administrative procedures, the strategic plan, the operational plan, among others, by the pattern of management through a model of process structure, which seeks to plan, define actions, metrics, controls, competencies, and strategic guidelines to obtain the goals of the organisation. The operation pattern provides the means through which the system flow works as a value chain that represents each one of the processes involved in the organisational transformation of organisations based on

the governance patterns framed in the models and standards of Innovation Management. Innovation and Digital Transformation are managed using this technology.

In the end, processes and operations generate business value, and with a continuous improvement approach, the operations procedures are continuously transformed, and the system is strengthened with the contribution that is developed through interaction in the model flow.

## 5. Discussion

The objective of this research was to identify elements, restrictions, and evidence of good practices based on the review of literature on models, standards, management, innovation, and transformation in areas related to governance and organisational management, with which a holistic and systematic structure was obtained that allowed the construction of a Conceptual Model of a Governance and Corporate Management System. The study also determined the key capabilities offered by the different support systems for model setup.

In addition, it became evident that sustainable Governance and Corporate Management Systems remain relatively understudied; therefore, this research also aims to close this gap by integrating Innovation Management models and standards into the proposed system to enable the responsible use of resources and ensure sustainable development of organisations.

The study incorporates the value of Technology Management and Innovation Management models and standards in the implementation and articulation of multidimensional dynamic processes focused on the development and execution of technology innovation plans, from which activities that generate new knowledge and strengthen the characteristics and capabilities of the organisation can maintain a culture of continuous and sustainable innovation.

To complement and strengthen the digital capabilities in the processes of the conceptual model of the Governance and Corporate Management System, Digital Transformation was added as a component. With the application of the strategies and technologies of this component, it is expected to create new links between organisations, new business models, and increase customer experience and value.

As can be seen, this study presents a new configuration of the business model, which was elaborated under the holistic approach of innovation processes and systems, with fundamental principles of transparency and integration. The governance pattern is the one that defines the functioning of each of the components, interactions, control, and direction of the organisation.

Surrounding the patterns of governance, management, and operation are the models and standards of Technology Management, Innovation Management, and Digital Transformation; these components generate a constant interaction, and as a consequence, a flow is generated in the system, which causes the frequent transformation of the business model, forcing the system to be in continuous evolution, allowing the reformulation of the strategic objectives, and, consequently, generating a transformation of the organisation.

The Governance and Corporate Management System model has been proposed as a value chain so that, in this way, strategic actions can be established in the organisational context, promoting the development of its strategic objectives, the fulfilment of its goals established in its indicators performance key and represented in each of the processes involved in the organisational transformation of companies. The model presented is simple and can be easily implemented in MSMEs. For this, it is necessary to select the KPIs of the companies; it is essential to consider various factors depending on the sector, such as size, structure, processes, complexity, KPIs, goods, and services, and whether it is a public, private, or mixed entity. Therefore, it is advisable to take into account the following factors.

Strategic alignment aims to align and understand the strategic objectives and key performance indicators of a company, with strategic objectives that can be structured from technology, innovation, and digital management. Different organisations have different purposes; some focus on operational excellence (quality), profitability (profit), and others on environmental, social, or political objectives, or are perhaps more balanced regarding

sustainability. Essentially, it is about ensuring that business objectives align with the goals derived from the strategic triad of Innovation, Digital Transformation, and Technology.

Risk management is crucial during any transition or transformation. It is essential to assess the existing level of risk and identify key indicators that require close monitoring. Focus on areas where the probability and impact of risk are high without ignoring the possibility of hidden risks that need to be managed.

Finally, a preliminary assessment is a helpful tool that helps identify the current status and maturity level of an organisation in implementing tripartite management. It also provides a snapshot of the levels of Technology management, Digital Transformation, and Innovation Management. By comparing it to the future state of the organisation, a gap can be established, which can help determine desirable indicators for the organisation. This information is crucial for developing a tripartite management implementation plan tailored to the unique characteristics of the organisation.

## 6. Conclusions

Previous studies have focused on analysing the influence of independent factors or components on corporate development. Consequently, this study can contribute to the literature with a new conceptual model that offers a novel perspective of integrating various factors or components and including sustainability guidelines in corporate governance to promote corporate sustainability performance.

In addition, there are various approaches to Innovation Management models and Technology Management models; a large number of the proposals focus on enabling the implementation of technology, but none implies the creation of a sustainable competitive advantage with a holistic and systematic structure that includes Governance, Innovation Management, and Technology Management together with Digital Transformation to develop capacities for strategic and sustainable innovation to generate Organisational Transformation.

In this study, as a contribution to solving this need, evidence is presented of a conceptual model that includes sustainability guidelines in the principle of transparency, which consists of an Open Governance and Corporate Management System with a dynamic perspective supported by the implementation of the processes mentioned above and obtaining, in the end, as a result of the interaction with the environment, the feedback of the Organisational Transformation process, which is vital to achieving sustainable development.

The system proposed in this study establishes a new configuration of the business model, which involves a regulatory foundation, the principle of transparency with guidelines for the management of sustainability and innovation, from which technology is managed, constituting a strategic element in which the organisation must interact with the environment through the capabilities offered by the different support systems and thus be able to obtain transparent sources of information that allow it to generate innovation and break down the barriers that prevent organisations from long-term sustainability.

Future work is expected to define the groups of organisational actors according to the capabilities that are part of the support systems to complement the model, to define the indicators that allow performance measurement, and to present a new structure of the proposed model involving the concept of Enterprise Architecture (EA). This new proposal will facilitate the implementation, verification, monitoring, and greater integration of the system in the organisation.

**Author Contributions:** A.G.-A. investigation, conceptualization, methodology, software, writing original draft preparation; W.N.B. validation, formal analysis, writing review and editing, supervision, funding acquisition; C.L.A. validation, formal analysis, supervision. All authors have read and agreed to the published version of the manuscript.

**Funding:** This research received no external funding.

**Data Availability Statement:** No additional data are available.

**Conflicts of Interest:** The authors declare no conflict of interest.

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
