# Peer review of "Governance and Corporate Management System Supported by Innovation, Technology, and Digital Transformation as a Driver of Change"

_sustainability, doi:10.3390/su151713150_

Round 1
Reviewer 1 Report
Article title: Conceptual Model for the structuring of a Corporate Governance and Management System supported in Innovation Management, Digital Transformation and Technology Management
After reviewing this paper carefully, I have some comments below:
- Abstract should be presented in a single paragraph. In addition, the abstract needs to be introduced through research results and implications.
- In the introduction, the authors need to clarify the new contribution of this study. Also, at the end of the introduction, the authors need to introduce the structure of this paper.
- This paper completely lacks a literature review section. This part is very important and the author must add it. In the literature review, authors must present relevant previous studies. In addition, the authors also need to update recent studies related to this topic. I suggest the authors review and cite related studies such as Almustafa et al. (2023); Nguyen and Dang (2023); Li et al. (2018); Nguyen and Dang (2022); …
- The discussion section should be related to the research context.
- There are many typos and grammatical errors. The authors need a full review.
References
Almustafa, H., Nguyen, Q. K., Liu, J., & Dang, V. C. (2023). The impact of COVID-19 on firm risk and performance in MENA countries: Does national governance quality matter? PloS one, 18(2), e0281148.
Li, J., Nan, L., & Zhao, R. (2018). Corporate governance roles of information quality and corporate takeovers. Review of Accounting Studies, 23(3), 1207-1240.
Nguyen, Q. K., & Dang, V. C. (2022). The Effect of FinTech Development on Financial Stability in an Emerging Market: The Role of Market Discipline. Research in Globalization, 100105.
Nguyen, Q. K., & Dang, V. C. (2023). The Impact of FinTech Development on Stock Price Crash Risk and the Role of Corporate Social Responsibility: Evidence from Vietnam. Business Strategy and Development. doi:https://doi.org/10.1002/bsd2.262
English is not good.
Author Response
Good morning dear reviewer, the adjustments made are detailed below:
The summary was corrected.
The introduction was corrected.
The literature review section was added.
The discussion and conclusions were adjusted to the context of the research.
The typographical and grammatical errors were corrected.
The adjustments were highlighted in the document.
Thank you very much for your willingness to review this document.

Reviewer 2 Report
This paper introduces a novel angle in the choice of research theme. Firstly, the paper comprehensively catalogues and scrutinizes literature pertinent to corporate governance, corporate management, technology management, innovation management, digital transformation, and organizational transformation, thereby extrapolating elements conducive to the construction of a conceptual model of corporate governance and management systems. Secondly, the article demarcates the conceptual model of corporate governance and management system structure into four sections: technology management model, innovation management model, digital transformation model, and organizational transformation model. Using extant research, the paper qualitatively analyses the role these four facets play in enhancing a company's capacity to adapt to market fluctuations, particularly the impact brought about by technology, innovation, digitization, and corporate competitiveness. The objective is to foster a more seamless linkage between scientific research, industry, and society. The innovative aspects of the paper are chiefly reflected in the analysis from the perspectives of technology, innovation, and digitization. Lastly, through the organization and analysis of the literature, this article proposes a governance and management model suitable for enterprises: the integration of technology management model, innovation management model, digital transformation model, and organizational transformation model to invigorate the corporate value chain. However, the paper does manifest significant issues that are recommended for the author's modification.
In the fourth section, the article posits that the model can be readily implemented in small and medium-sized enterprises, but fails to substantiate this claim. Furthermore, there is a lack of detailed discussion on the role of the newly proposed business model in enterprise development.
Enterprises, each at different stages of development and facing varying market pressures, have diverse needs that necessitate refined adjustments in their governance and management systems. The paper only offers a generalized analysis and proposes a new business model configuration. It is recommended that the paper categorizes enterprises, and specifically analyses how different systems in the new business model offer varying capabilities to assist companies in making improvements, with the aim of establishing a more reasonable governance and management system. Besides, while the paper builds a conceptual model of innovation management, digital transformation, and technology management supporting corporate governance and management systems based on the summary of existing literature, the model is untested in practice or experience and can only be deemed as a model hypothesis or theoretical assumption. The practicality and effectiveness of the model require validation in practice. It is suggested that the article be supplemented with case studies or data for verification.
Author Response
Good morning dear reviewer, the following are the adjustments made:
The discussion was adjusted with the context of the research.
The conceptual model proposed in this article is the first part of my doctoral research; it was empirically tested with a bibliometric analysis, an expert judgment and through an Exploratory Factor Analysis (EFA) and a pilot test that part of the research is part of another article that is also in the process of publication.
The adjustments were highlighted in the document.
Thank you very much for your willingness to review this document.

Reviewer 3 Report
The paper although interesting needs more work as this is a conceptual paper thus needs more justification and elaboration.
1. The abstract should be in one paragraph.
2. The abstract should also include some implications.
3. The authors need to employ some protocol like PRISMA for the review.
4. Inclusion, exclusion, coverage etc need to be clearly detailed.
5. What was found interesting and what is the way forward for research and practice needs more work.
6. Comprehensiveness and currency of the literature needs to be checked.
Acceptable.
Author Response
Good morning dear reviewer, the adjustments made are detailed below:
The abstract was corrected.
The conceptual model proposed in this article is the first part of doctoral research, the PRISMA protocol was not used, but it was empirically tested with a bibliometric analysis, an expert judgment and an Exploratory Factor Analysis (EFA); that part of the research is part of another article that is also in the process of publication.
The process of literature collection was detailed.
The adjustments were highlighted in the document.
Thank you very much for your willingness to review this document.

Reviewer 4 Report
1. the article is written on the current topic
2. insufficient analytics of publications that have researched this topic
3. models of different countries are not well described, with an emphasis on advantages and disadvantages
Author Response
Good morning dear reviewer, the adjustments made are detailed below:
The collection of documents, the use of literature, and the methods used in their analysis were detailed.
The literature review section was added, and the governance models were added with emphasis on the advantages and disadvantages.
The adjustments were highlighted in the document.
Thank you very much for your willingness to review this document.

Round 2
Reviewer 1 Report
I am satisfied with this version. It can be published.
English is ok
Author Response
Thank you very much for your contributions. Greetings.
Reviewer 2 Report
1. Discussion on easy implementation in MSMEs.
The paper holds that this model promotes the continuous development of management system in the interaction of technology management, innovation management, digital transformation and organizational transformation, so as to frequently update the business model of enterprises, and ultimately improve the competitiveness and sustainable development ability of enterprises. In this process, enterprises need to develop a lot of strategic goals, and the realization of these strategic goals will be reflected in the key performance indicators of enterprises and organizational transformation. The paper may think that these key performance indicators are easy to obtain, so this model is relatively easy to achieve for small and medium-sized enterprises. There is no problem with the logic of such an argument, but for an enterprise, there are many indicators that are very important, and it is better to choose which indicator as the criterion for judgment. And as the review comments said, for different industries, different periods, different types of enterprises, the development focus and the external environment are not the same, at this time, how should we choose indicators? This all requires further clarification.
2. The role of the model for enterprise development
In this article, the author has carried on a detailed theoretical discussion from the sustainable development ability and competitiveness of enterprises. The article believes that it combines the concepts of different management fields such as technology management, innovation management, digital transformation and organizational transformation, and helps enterprises optimize resource allocation and constantly update business models to better adapt to changes in the global market through the interaction of various fields. At the same time, the integration of these areas can help enterprises improve technology utilization, promote technological innovation and digital transformation, help enterprises respond to changing customer needs, and improve market competitiveness. However, the requirements of reviewers have not been fully realized, and there is no demonstration with examples or data. In this case, the effect of the model can only stay at the theoretical level.
3.The citation of references is a bit excessive in some places, for example, the first sentence of the article cites 11 references. The title of the article is too long.
Author Response
Good morning dear reviewer, the following are the adjustments made:
- Recommendations were added for choosing indicators depending on the sector (Discussion).
- The result of a system validation technique was added. It is not prudent to add the results of the validation of the system using Exploratory Factor Analysis (EFA) and Confirmatory Factor Analysis (CFA) as these results are part of another article that is in the process of publication.
- The citations in various parts of the article were better organised, and the title was shortened. The adjustments were highlighted in the document.
Thank you very much for your willingness to review this document.
